# Study of Bond Formation in Ceramic and Composite Materials Ultrasonically Soldered with Bi–Ag–Mg-Type Solder

**DOI:** 10.3390/ma16082991

**Published:** 2023-04-10

**Authors:** Roman Kolenak, Tomas Melus, Jaromir Drapala, Peter Gogola, Matej Pasak

**Affiliations:** 1Faculty of Materials Science and Technology in Trnava, Slovak University of Technology in Bratislava, Jána Bottu No. 2781/25, 917 24 Trnava, Slovakia; 2Department of Non-Ferrous Metals, Refining and Recycling, FMT-Faculty of Materials Science and Technology, Technical University of Ostrava, 17. Listopadu 15, 708 33 Ostrava, Czech Republic

**Keywords:** soldering, Al_2_O_3_ ceramic, Ni–SiC substrate, Bi solder, ultrasonic soldering

## Abstract

This research aimed to study a Bi–Ag–Mg soldering alloy and the direct soldering of Al_2_O_3_ ceramics and Ni–SiC composites. Bi11Ag1Mg solder has a broad melting interval, which mainly depends on the silver and magnesium content. The solder starts to melt at a temperature of 264 °C. Full fusion terminates at a temperature of 380 °C. The microstructure of the solder is formed by a bismuth matrix. The matrix contains segregated silver crystals and an Ag (Mg, Bi) phase. The average tensile strength of solder is 26.7 MPa. The boundary of the Al_2_O_3_/Bi11Ag1Mg joint is formed by the reaction of magnesium, which segregates in the vicinity of a boundary with a ceramic substrate. The thickness of the high-Mg reaction layer at the interface with the ceramic material was approximately 2 μm. The bond at the boundary of the Bi11Ag1Mg/Ni–SiC joint was formed due to the high silver content. At the boundary, there were also high contents of Bi and Ni, which suggests that there is a NiBi_3_ phase. The average shear strength of the combined Al_2_O_3_/Ni–SiC joint with Bi11Ag1Mg solder is 27 MPa.

## 1. Introduction

Over the past 25 years, new electronic applications have led to incredible changes in how power equipment is packaged. These products, which are destined for home users, typically include handheld devices and appliances for domestic use. As an example, we may mention gaming consoles, radio receivers, or bigger devices such as TV sets and intelligent appliances. The trend in these devices is for them to be smaller, faster, and cheaper, as well as produced in large quantities. Their applications include a diverse range of technologies, such as digital microprocessors, (built-in and stand-alone) memories, devices with wireless functions, and many others [1,2,3]. In many cases, a packaged device forms a part of an electronic system and determines its output power. The trend in packaging for these applications consists of switching to modules, which may be easier to integrate and form the system with a complex package (i.e., the multi-chips and passive devices) [4,5,6,7].

For integrated electronic components, the interconnect material is solder. It is used for joining the packages and/or modules with the motherboards. The soldering alloys are also used for connecting the semiconductor matrix with the package by means of die-attach and/or flip-chip interconnections [8,9,10,11,12]. The solder acts not only as the electric conductor but also as the thermal conductor, and it mechanically holds the parts together in the correct position. Thus, it should exert good electric and thermal properties, which predestine it for such applications. Moreover, it must have a relatively low melting point and the capability to wet the materials used. There are often combinations that necessitate joining the conductive or semiconductor parts with the passive substrates, which do not transmit the electric current. In the case of assembling a packaged part, this alloy must have a lower melting point than the other solders used in the package in order not to damage the device as a whole. For such applications, it is advisable to use the solders based on Sn–Cu, Sn–In, Sn–Sb, Sn–Bi, or Bi–Sn [12,13,14,15,16,17].

The present trend also includes using suitable technologies for creating the joint between the mentioned materials. In recent years, it has become more popular to apply the technologies of joint fabrication without the use of flux or interlayers of coatings, which ensure the wetting of hard-to-wet materials. The hard-to-wet materials used for the electronic packaged components are, for example, Al_2_O_3_, AlN, or Si_3_N_4_. Ultrasonic power is one of the ways that these materials can be directly soldered together. This method is becoming more popular, owing to the increased speed of the process as well as time and financial savings for the producers. This method is increasingly used mainly because of the speed of the process and the time and financial savings it provides manufacturers. The condition for wetting is then not only the ultrasound itself but also a suitably designed solder alloy containing an active element to ensure wetting. Such an alloy is then called an “active solder” [18,19,20,21,22,23].

The most commonly used active element in solders is Ti. The effects of solders alloyed with titanium are documented in the research works of many authors [23,24,25]. The Sn–Ag–Ti-based solder allowed for the fabrication of the joints with the substrates of graphite, SiC, AlN, 1Cr18Ni9Ti steel, or Al_2_O_3_ ceramics. By application of ultrasonic power, the bond with a high concentration of Ti at the joint boundary was formed in all cases. Titanium is an element with a high affinity for the Al, O, Si, C, and N elements. It forms the following compounds with these elements: AlTi, Al_2_Ti, TiO_2_, TiSi_2_, TiC, and TiN. These compounds then ensure the wetting of substrates and bond formation [26,27].

In our previous studies [28,29], we have devoted attention to the characterization of In–Ag–Ti-based solder and the soldering of SiC ceramics. In that case, the effect of Ti on joint formation was not proven. The joint boundary was formed by a high concentration of In. The joint has exerted an adhesive character, and the compounds Ti–C and/or Ti–Si were not observed. On the other hand, in the case of soldering Si with Sn–Ag–Ti solder, the reaction layer of TiSi_2_ was clearly proven.

Another field of our research was the study of the applicability of Bi-based solders. In work [30], we devoted ourselves to the study of direct soldering of Al_2_O_3_ ceramics with Cu by application of Bi11Ag2La solder. Lanthanum was added to the solder as an active element to ensure the wetting of the Al_2_O_3_ surface. A thin layer of La, 1 μm in thickness, was observed. The strength of the Al_2_O_3_ joint attained a value of 20 MPa.

A study by the authors [31] investigated the interfacial reaction between Cu substrates and Bi–Ag solder. The stoichiometric Cu–Bi phase that formed isothermally in the liquid solder significantly affected the Cu dissolution. The results also show that the Bi–Ag/Cu joints had better shear strength than the Pb–Sn/Cu joints, indicating that the mechanical bond at the grain boundary was strong enough to resist shear deformation.

The authors [32] also conducted research on the microstructure and properties of the Bi–Ag alloy. This gold exhibited non-uniform solidification and significant undercooling. When the Ag content was increased, the tensile strength was effectively increased.

Another study investigated a bismuth-based, germanium-doped alloy with a composition of Bi-11Ag-0.05Ge that is suitable as a replacement for high-Pb solder [33]. This alloy has a solidus at 262.5 °C and a liquidus at 360 °C. The alloy has a shear modulus of 13.28 GPa and a tensile strength of 59 MPa. An element such as Ge can be added in small amounts (≤500 ppm) to improve wetting. Although this investigated solder exhibits lower thermal conductivity and poorer wetting properties, it has a lower shear modulus than current solder and a higher tensile strength than Pb-5Sn solder. These properties allow it to exhibit high thermomechanical fatigue resistance.

In this article [34], a new high-temperature solderable gold Bi-11Ag with Sn addition of 1.3–5 wt.% was developed. The results show that the melting temperature of Bi-11Ag solder dropped from 265 to 255 °C after the addition of Sn. The microstructure was refined when the amount of Sn was less than 3 wt.% but coarsened with the increase of Sn. The wettability and tensile strength of the Cu/Bi-11Ag-xSn solder/Cu solder joint improved with increasing the amount of Sn. The tensile strength of the Cu/Bi-11Ag-5Sn/Cu solder joint is higher than that of the Cu/Pb-5Sn/Cu joint due to the formation of the Ag_3_Sn layer.

Therefore, this work deals with the study of Bi–Ag–Mg type soldering alloy and the research of ultrasonic soldering of Al_2_O_3_ ceramic and Ni–SiC composite material.

The solder type Bi–Ag–Mg is destined for higher application temperatures. Magnesium in solder was selected as an active element for wetting the ceramic and composite Ni–SiC substrates. The solder is alloyed with silver to increase its strength and electric conductivity. Thus, the research comprises an evaluation of the solder itself, followed by an investigation of interaction at the solder/substrate interface during hot plate/ultrasound soldering.

## 2. Experimental

The joints were fabricated by using an active solder type Bi–Ag–Mg, whose chemical composition is given in Table 1. The chemical analysis of the alloy was performed by the method of atomic emission spectrometry with induction-coupled plasma (ICP-AES). The analysis of individual solder elements was carried out on the SPECTRO VISION EOP. The alloy samples were dissolved in suitable chemical solutions of acids and bases for ICP-AES analysis. The analysis itself was carried out on an atomic emission spectrometer with a pneumatic atomizer and Scott’s sputtering chamber.

Three test pieces for the tensile strength measurement of solder were prepared, and their dimensions are presented in Figure 1. All values are given in mm. The thickness of the test piece was 4 mm. The tensile strength was tested on a LabTest 5.250SP1-VM. The loading rate of the specimen is 1 mm/min.

The application of Bi11Ag1Mg solder for experimental soldering was carried out using the following substrates: Ni–SiC composite in the form of a disk with a diameter of Φ 15 mm × 3 mm,Al_2_O_3_ ceramics in the form of a disk with a diameter of Φ 15 mm × 3 mm,Ni–SiC composite in the form of a square with dimensions of 10 × 10 × 3 mm.

The scheme of the soldered joint, prepared for the chemical analysis of substrate/solder/substrate boundaries and measurement of shear strength, is shown in Figure 2.

A hybrid hot-plate/ultrasonic soldering method was used to produce the joints. The soldering temperature was adjusted using a thermostatic control on the heating element. A substrate of each material was placed on the hot plate, and the solder was placed on top of it. The materials thus deposited were heated to soldering temperature. After the solder melted on the surface of the substrates, ultrasonic energy was introduced into the liquid solder. A HANUZ UT2 type device was used to generate the ultrasonic energy. The transfer of ultrasonic energy to the molten solder was provided by an encapsulated ultrasonic transducer with a piezoelectric oscillation system and a titanium sonotrode with a tip diameter of Ø 3 mm. A schematic representation of the application of ultrasonic vibration to the molten solder is shown in Figure 3. Figure 4 shows a schematic of the ultrasonic brazing process.

The temperature during soldering was continually controlled by the use of a NiCr–NiSi thermocouple. Owing to the ultrasound’s presence and its cavitation and cleaning effect, the flux was unnecessary. After ultrasound activation, the redundant solder was removed, and the individual couples of substrates were laid on top of each other, and the joint was thus formed. The soldering parameters are given in Table 2.

The manufactured samples were processed using standard metallographic techniques of sanding, burnishing, and pickling to accentuate the microstructure. In particular, the aim was to highlight the transition regions at the interface between the solder and the substrate. Grinding was performed on the SiC grinding disks with grain sizes of 600, 1200, and 2400 grains/cm^2^.

The polishing was carried out using a diamond emulsion with diamond particle sizes of 6 μm, 3 μm, and 1 μm. Polishing was realized on the polishing disks from the BUEHLER company. After the polishing of the specimens, the etching process followed. The etching type HCl: HNO_3_ with a 1:3 concentration was used for 1 s. The analysis of solder microstructure was performed by scanning electron microscopy (SEM) on the equipment type TESCAN VEGA 3.

XRD analysis was performed on metal filings from cast and annealed samples using a PANalytical Empyrean X-ray diffractometer (XRD) (Malvern Panalytical Ltd., Malvern, UK). The measurement procedure on metal filings instead of bulk castings was chosen to limit the influence of casting texture on the recorded XRD pattern. The measurements were performed in Bragg–Brentano geometry. Theta–2theta angle range between 20° and 145° 2theta was chosen. The XRD source was a Co-anode lamp set to 40 kV and 40 mA. The incident beam was modified by a 0.04 rad Soller slit, 1/4° divergence slit, and 1/2° anti-scatter slit. The diffracted beam path was equipped with a 1/2° anti-scatter slit, 0.04 rad Soller slit, Fe beta filter, and PIXcel3D position-sensitive detector operating in 1D scanning mode. The phase quality was analyzed using the PANalytical Xpert High Score program (HighScore Plus version 3.0.5) with the ICSD FIZ Karlsruhe database.

The chemical element analysis was performed via the Oxford Instruments X-Max silicon drift detector and energy dispersive X-ray spectrometer (EDS, Oxford Instruments plc, Abingdon, UK).

The DTA analysis of the Bi11Ag1Mg solder was performed on the equipment type DTA SETARAM Setsys 18TM. The heating rate of the specimen was 5 °C/min from room temperature until the complete fusion of the specimen. The results included the temperatures of phase transformations in the liquidus—solidus range and phase transformations in the solid state, and the enthalpies of phase transformations were also determined.

Due to different coefficients of thermal expansivity of materials used, the soldered joints are during service loaded mainly in shear. Therefore, the shear strength measurements were performed with equipment type Lab-Test 5.250SP1-VM. These measurements were performed with the use of a special jig for positioning specimens to ensure uniform shear loading. The schematic representation of this setup is shown in Figure 5.

## 3. Experimental Results

### 3.1. DTA Solder Analysis

The DTA analysis was always performed twice at the heating and cooling rates of 5 °C/min (Figure 6 and Figure 7). The significant phase transformation temperatures in the Bi11Ag1Mg type solder alloy determined by TG/DTA analysis are documented in Table 3. The main aim of the DTA analysis was to determine the melting interval. It resulted in a DTA curve, which is actually the graph of thermal flow. The phase containing 49 at.% Ag, 25 at.% Mg, and 26 at.% was primarily segregated in the solidification process, which corresponded to the Ag (Mg, Bi) formula. From an ICP-AES analysis, it was found that the total average concentration of silver in the specimen was 12.2 wt.%, corresponding to 21.2 at.% Ag. The blue lines in Figure 8 show where this composition fits, which suggests that the liquidus temperature given by the Ag–Bi phase diagram exactly matches the temperature found by the DTA analysis. In the next phase of solidification, the melt was enriched with bismuth, whereby the crystals rich in Ag have grown to the eutectic point at 263 °C and formed the fine lamellae of the (Ag) + (Bi) matrix. Mg was not observed at the eutectic point. In the cooling curves, two thermal effects again occurred that were not observed during the cooling phase. It may be related to the existence of two reactions, namely the primary Ag (Mg, Bi) phase and the formation of a solid solution (Ag) enriched by magnesium and in a lesser measure by bismuth in the next phase.

### 3.2. Microstructure of the Solder Bi11Ag1Mg

The microstructure of the soldering alloy type Bi11Ag1Mg is formed of a bismuth matrix. In the solder matrix, the crystals of a solid solution of silver (Ag) and intermetallic phases of magnesium are distributed. The microstructure with the designation of phases is documented in Figure 9. The analysis in spectra 1 and 2 contained 25.4 at.% Mg, 26.9 at.% Bi, and 47.7 at.% Ag (large grey areas). The binary Ag–Bi system was of the eutectic type, with minimum solubility of both components in the matrix. The Bi–Mg system contained a stable intermetallic phase of α-Mg_3_Bi_2_ [36], and the AgMg phase was observed in the Ag–Mg system over a relatively broad range of concentrations [37]. It is probable that the substitution of Mg atoms with Bi atoms took place. Thus, it can be concluded that this phase can be stoichiometrically described by the formula Ag (Mg, Bi), where the proportion of Mg:Bi atoms is equal to 1:1.

The small, darker constituents of irregular shape (spectra 3 and 4) contained the same elements but in different proportions, namely 11.4 at.% Mg, 1 at.% Bi and 87.6 at.% Ag. The intermetallic compound type Ag_3_Mg is thus concerned there, with all probability occurring over a relatively broad range of concentrations (Figure 10) [18].

Ag and Bi were also observed in spectra 5 and 6. These components may form a two-phase zone with a eutectic mixture of Bi + (Ag). The bright zones of the matrix contain pure bismuth (spectra 7 and 8).

XRD analysis of Bi11Ag1Mg solder showed the presence of a AgMg type intermetallic phase, solid solution (Ag), pure bismuth, and magnesium.

The results of the diffraction analysis are shown in Figure 11.

### 3.3. Tensile Strength of the Solder Alloy

Mechanical tests were carried out to measure the tensile strength of the Bi11Ag1Mg active soldering alloy. The dimensions of the test pieces were drafted and calculated. Three specimens were used for tensile strength measurements of the experimental solder, and the loading rate of each specimen was 1 mm/min. From the measured values, an average tensile strength of 26.67 MPa was calculated, and the results for the measured values are shown in Figure 12. The ductility of the test pieces for the Bi11Ag1Mg alloy varied from 0.58% to 1.92%. The tensile strength curve of the experimental solder alloy is shown in Figure 13. The fixing points of the origin are indicated by the colour blue. The gut color indicates the force on the contracted yield strength. The green colour indicates the maximum load (force at the interface).

For the comparison of Bi11Ag1Mg solder, the measurement was also performed on other alloys of type Bi11Ag3Ti and Bi11Ag1.5Ti1Mg to enable the comparison of the effect of adding an element on the resultant strength of the experimental alloy.

From the results of tensile test measurements shown in Figure 12, it is obvious that the highest strength values were achieved with the solder type Bi11Ag1.5Ti1Mg, in which the highest average tensile strength of 39 MPa was measured. With the addition of just 1.5 wt.% Ti to the solder alloy type Bi–Ag–Mg, the average tensile strength increased by almost 20%. Titanium is an active element that reinforces the solder matrix. The lowest values of tensile strength were achieved with the solder type Bi11Ag3Ti, namely an average tensile strength of 26 MPa. Therefore, it should be accepted that the solder type Bi11Ag3Ti exerts similar tensile strength to the analyzed solder type Bi11Ag1Mg.

### 3.4. Analysis of the Transition Zone in the Al_2_O_3_/Bi11Ag1Mg Joint

To determine the chemical composition and identify the individual phases, EDX analysis was performed in the solder joint (Figure 14). The spectra 1 to 4 show the scatter in the concentration of individual elements present at a given location. The average Bi content is 10.4 at.%, Ag 18.7 at.%, Mg 2.0 at.%, Al 59.9 at.%, and O 11.5 at.%. These data show that the Al:O proportion no longer corresponds to the stoichiometric proportion of 2:3. However, the transition between the ceramic and soldered zones is continuous and is related to the diffusion processes between Al_2_O_3_ and Bi–Ag–Mg solder. Mainly silver and magnesium have segregated in the boundary vicinity, and these elements have interacted with aluminium. However, a new phase was not formed, only a mixture of individual metals, mostly as their solid solutions in different mutual proportions.

The zone is closest to the interface between the Al_2_O_3_ substrate and solder (spectrum 5). Occurrence of a solid solution of (Bi) + Al_2_O_3_.

Soldered zone (planar analysis)—occurrence of actually pure Bi, partial diffusion of aluminium, and oxygen was not identified (spectrum 6).

In spectrum 7, the Al_2_O_3_ substrate was identified.

In the binary Ag–Bi and Al–Bi systems, no intermetallic compounds were observed. In the binary Mg–Bi [36] system, a high-temperature intermetallic phase type Mg_3_Bi_2_ exists with a melting point of approximately 823 °C, which, however, was not formed in this case.

The planar distribution of elements on the boundary is documented in Figure 15. From the planar distribution, it is obvious that magnesium significantly contributes to bond formation. The green color identifies the magnesium presence on the boundary. Magnesium is also partially diffused into the substrate zone.

The line analysis in Figure 16 presents the distribution of elements in the studied part of the soldered joint of Al_2_O_3_/Bi11Ag1Mg. Mainly, magnesium was segregated on the joint boundary. The thickness of the reactionary Mg layer in this joint was approximately 2 μm.

### 3.5. Analysis of Transition Zone in Bi11Ag1Mg/Ni–SiC Joint

For determining the chemical composition and defining the individual phases, the EDX analysis of the soldered joint was performed (Figure 17). Measurement was performed in eight points, namely spectra 1 to 8.

In measurement point spectrum 1, the analysis of the Ni–SiC substrate zone in the form of eutectics was performed (carbon cannot be analyzed by EDX analysis). The Ni:Si proportion is approximately 75:25, which stoichiometrically corresponds to the β1-Ni_3_Si phase, as proved also by the binary diagram of the Ni–Si system [38]. The smaller, darker constituents probably contain carbon.

In measurement point spectrum 2, a high Ag content (42 at.%), then Mg (2.3 at.%), and Bi (7.7 at.%) were observed. Moreover, a high Ni (41 at.%) and Si (7 at.%) content was also detected. Both of these elements originate from the substrate. Ag and Ni elements are mutually insoluble.

In measurement point spectrum 3, a high Bi (87 at.%) and Ni (11.8 at.%) content was measured, which may suggest the existence of the NiBi_3_ phase that was identified in the binary diagram of Bi–Ni [39] (stoichiometric proportion of Bi:Ni = 75:25). A higher Bi content suggests the pickup of pure Bi matrices during analysis.

At the measurement points for spectra 4 and 7, a solid solution was observed with a high Bi content in the matrix and a low Ag content (below 1 at.%). In one case, Mg was also observed in the bismuth matrix. Si and Ni elements coming from the Ni–SiC substrate were identified. This effect was primarily caused by ultrasonic soldering and secondarily by diffusion. The formation of other phases was not observed in these zones. The bond between the Ni–SiC substrate and the solder was planar, sound, and without visible defects. This can be attributed to the favorable effects of Bi on the interaction between the solder and the substrate.

A planar analysis of the bright grey zone (spectrum 5) showed a high content of Ag (45.5 at.%), Mg (30.8 at.%), and Bi (21.2 at.%). The binary Ag–Bi system was of the eutectic type, with minimum solubility of both components in the matrix. A stable intermetallic phase of α-Mg3Bi2 was detected in the Bi–Mg system, and the AgMg phase was observed in the Ag–Mg system, over a relatively broad range of concentrations. Regarding the existence of the AgMg phase, it is probable that the substitution of Mg atoms with Bi atoms took place, which suggests from a stoichiometric perspective that this phase can be described as Ag(Mg,Bi) [36,37]. The proportions of these atoms may be slightly affected by the presence of Ni (1.9 at.%) and Si (0.6 at.%) in the substrate. 

A local analysis of Spectrum 6 revealed the presence of Bi (75.8 at.%) with high proportions of Ag (18.3 at.%) and Mg (3.8 at.%). It concerns the mixture of fundamental elements, which is formed by the ternary eutectic reaction as the two-phase and/or three-phase zones. This region is very heterogeneous. 

The smaller dark zones in Spectrum 8 contained approximately 87.8 at.% Ag, 8.4 at.% Mg and 3 at.% Bi. In all probability, this indicates the intermetallic phase of Ag_3_Mg over a relatively broad concentration range.

The results of a planar EDX analysis show the distribution of elements at the boundary, as presented in Figure 18. The transition zone of the Bi11Ag1Mg/Ni–SiC joint was analyzed, in which the interaction between the molten bismuth solder and the Ni–SiC substrate took place. The zone between the boundary and the solder has a uniform composition and is formed mainly of silver phases.

The line analyses in Figure 19 show the concentration profile through the Ni–SiC/solder boundary over a total length of 10 μm. The transition zone consists of Bi and Ag elements, as proved by the line profiles. The bond was formed due to the increased Ag concentration, as is obvious from Figure 18. The thickness of the reaction Ag phase at the boundary of this joint was approximately 1 μm.

### 3.6. Shear Strength of the Soldered Joints

The research in this work was oriented toward soldering Al_2_O_3_ ceramics and Ni–SiC composites. This study aimed to assess the suitability of the newly developed solder type, Bi–Ag–Mg. Figure 20 shows that when Al_2_O_3_ ceramics and Ni–SiC composite were joined together with Bi11Ag1Mg solder, the average shear strength of the joint reached 27 MPa. Figure 20 shows that the highest values of shear strength were achieved in the case of an Al_2_O_3_/Ni–SiC joint fabricated by using Bi11Ag1.5Ti1Mg solder, in which the highest shear strength of 54 MPa was observed. The lowest scatter was found in the case of the Al_2_O_3_/Ni–SiC joint fabricated by using Bi11Ag3Ti solder, in which the lowest shear strength of 23.5 MPa was measured. With the addition of a small amount of Ti (1.5 wt.%) to solder the alloy, its average shear strength increased by 50%. Titanium is an active element that reinforces the solder matrix.

For a more precise identification of the mechanism of bond formation, the fractured surfaces of soldered joints were analyzed. Figure 21 shows the fractured surface of the boundary in the Al_2_O_3_/Bi11Ag1Mg/Ni–SiC joint. It can be seen that the fractured surface remained covered with solder from the silicon side. We suppose that a ductile fracture has occurred in the solder. The analysis of the planar distribution of Al, O, Ni, Si, Mg, Ag, and Bi elements on the fractured surface was performed, as documented in Figure 21b–h. In the planar distribution of Al and O elements, which in Figure 21b,c represents the base metal, one may see the local zones where the tearing off of solder from the substrate has occurred. The planar distribution of Ag and Mg elements on the substrate is documented in Figure 21g,h, where it is evident that magnesium and silver have segregated on the surface of the Al_2_O_3_ substrate. Figure 21h shows the planar distribution of Mg on an Al substrate. It is obvious that Mg is segregated on the surface of the Al substrate.

## 4. Conclusions

This research aimed to characterize the Bi–Ag–Mg type of soldering alloy. The proposed composition of the soldering alloy was studied for its suitability for soldering Al_2_O_3_ ceramics and Ni–SiC composites through the application of ultrasonic soldering. The following results were achieved:The melting point of Bi11Ag1Mg solder and its reactions during heating were determined using DTA analysis. During solidification, the melt was enriched by bismuth, while Ag-rich crystals were grown up to the eutectic point. At 264 °C, fine lamellae of (Ag) + (Bi) matrix were formed. Two thermal effects occurred during cooling, which may be related to the existence of two reactions involving the primary Ag (Mg, Bi) phase and a solid solution (Ag) enriched by magnesium and bismuth.The mechanical test determined the tensile strength of the active soldering alloy type Bi11Ag1Mg. The average tensile strength of Bi11Ag1Mg solder is 26.7 MPa. When only 1 wt.% Mg is added, the tensile strength of the solder is partially increased.On the boundary of the Al_2_O_3_/Bi11Ag1Mg joint, a reaction layer that was mostly formed of Mg was identified. This thin layer was approximately 2 µm in thickness, and it was responsible for bond formation with the surface of Al_2_O_3_ ceramics. It is evident from the areal distribution that magnesium contributes significantly to the formation of the joint. Magnesium partially diffused into the substrate zone.Bi, Ag, and Mg formed the majority of the bond in the boundary between the Bi11Ag1Mg/Ni–SiC joint. The interface was found to have high Ni and Si contents, which originated from the substrate. The high content of Bi and Ni was found by experiment, which may indicate the existence of the NiBi3 phase. The region between Bi11Ag1Mg/Ni–SiC is mainly uniformly formed by silver phases. The thickness of the reactive silver phase at the interface was approximately 1 μm. These reaction phases located at the interface are responsible for the formation of the junction with the Ni–SiC surface.Research on soldering the combination of Al_2_O_3_ ceramics and Ni–SiC composites by use of Bi11Ag1Mg solder has proven the suitability of the selected solder alloy. The average shear strength of the Al_2_O_3_/Bi11Ag1Mg/Ni–SiC joint has reached 27 MPa.

## Figures and Tables

**Figure 1 materials-16-02991-f001:**
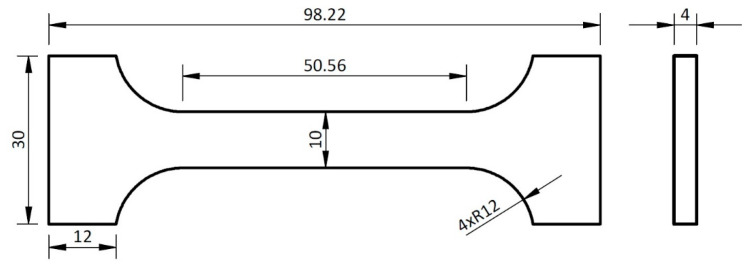
Solder test body for static tensile strength test.

**Figure 2 materials-16-02991-f002:**
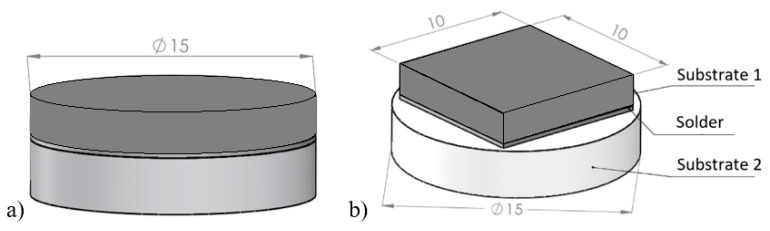
Setup of the soldered joint, (**a**) for analysis of substrate/solder/substrate boundaries, and (**b**) for shear strength measurement.

**Figure 3 materials-16-02991-f003:**
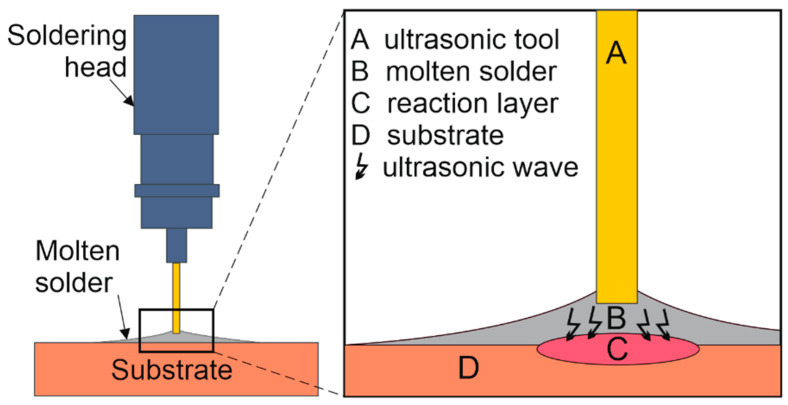
Schematic illustration of ultrasonic activation on the substrate surface.

**Figure 4 materials-16-02991-f004:**
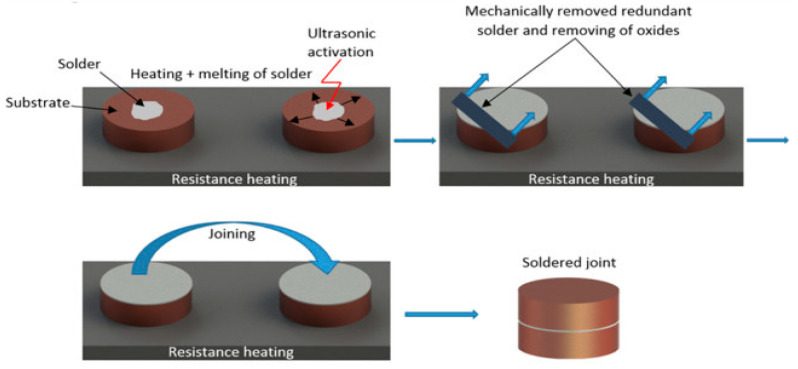
Schematic representation of ultrasonic soldering.

**Figure 5 materials-16-02991-f005:**
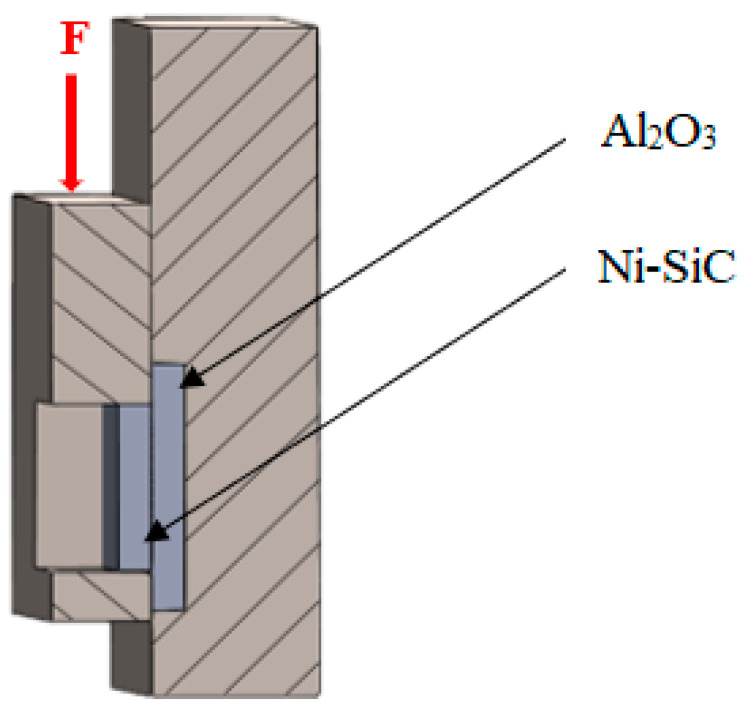
Schematic representation of a jig for shear strength measurement.

**Figure 6 materials-16-02991-f006:**
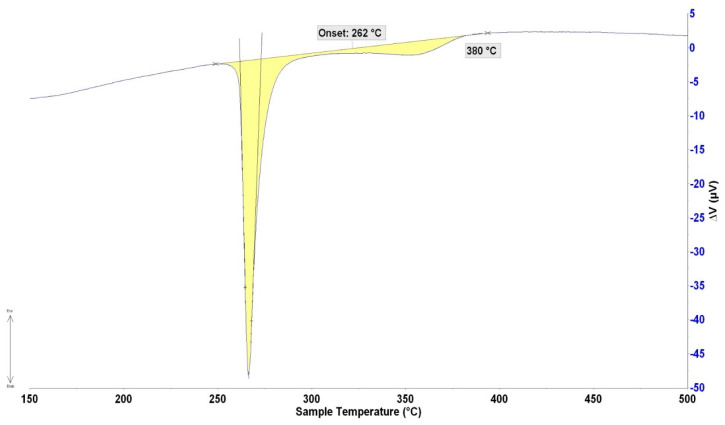
DTA analysis of Bi11Ag1Mg solder, the heating rate of 5 °C/min.

**Figure 7 materials-16-02991-f007:**
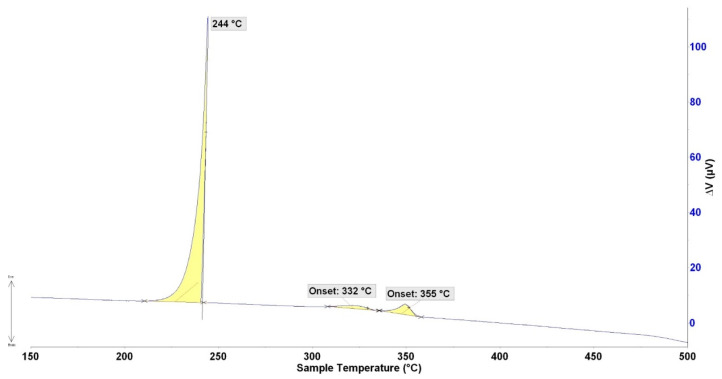
DTA analysis of Bi11Ag1Mg solder, the cooling rate of 5 °C/min.

**Figure 8 materials-16-02991-f008:**
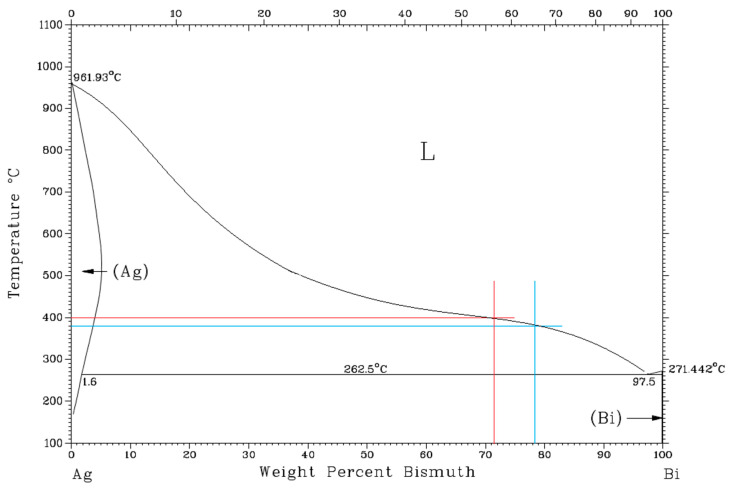
Binary diagram of silver–bismuth [35].

**Figure 9 materials-16-02991-f009:**
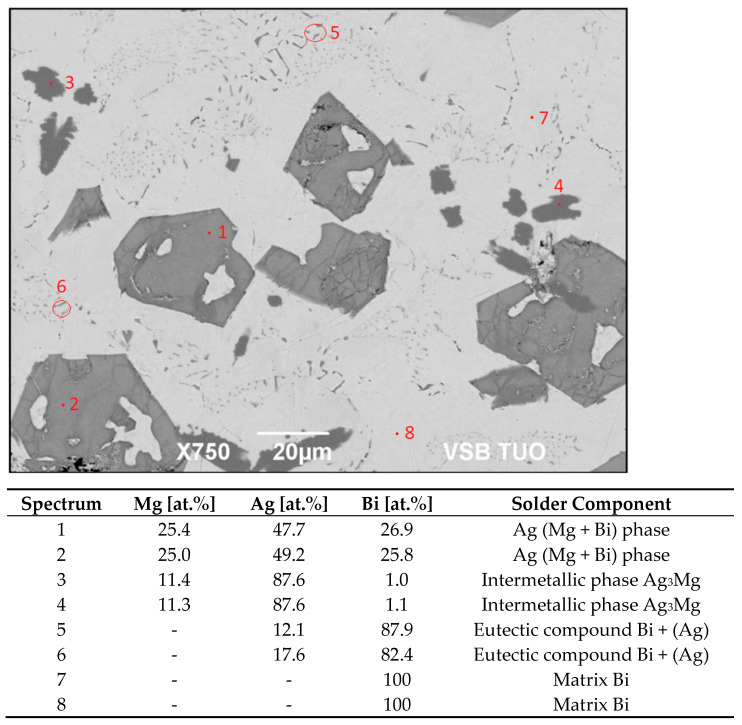
Microstructure of the Bi11Ag1Mg solder.

**Figure 10 materials-16-02991-f010:**
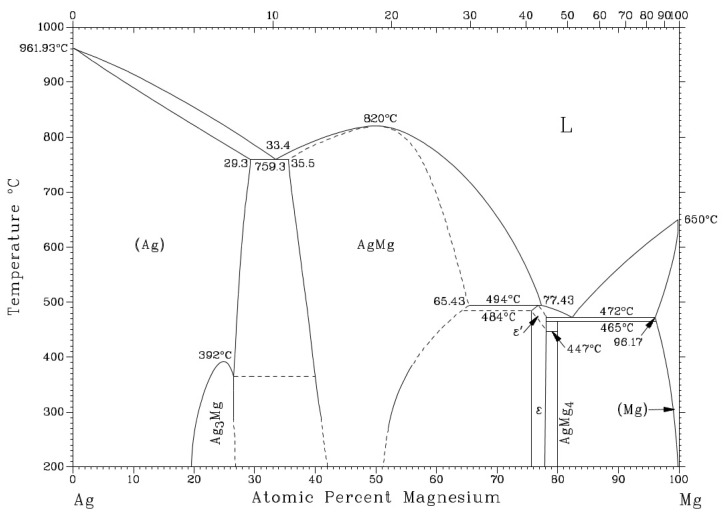
Binary diagram of silver–magnesium [37].

**Figure 11 materials-16-02991-f011:**
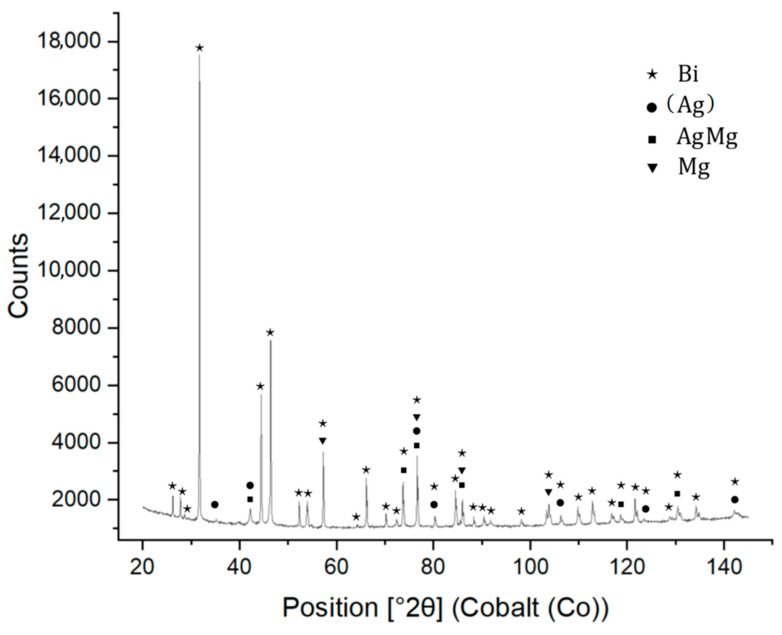
XRD analysis of Bi11Ag1Mg solder.

**Figure 12 materials-16-02991-f012:**
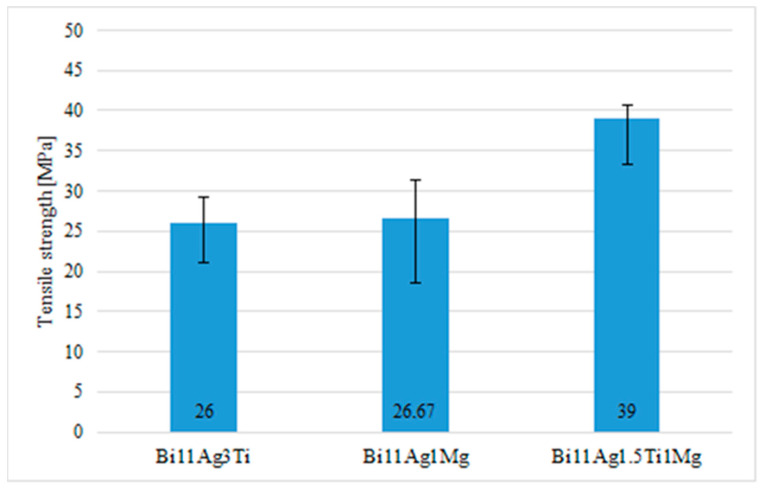
Tensile strength of experimental soldering alloys.

**Figure 13 materials-16-02991-f013:**
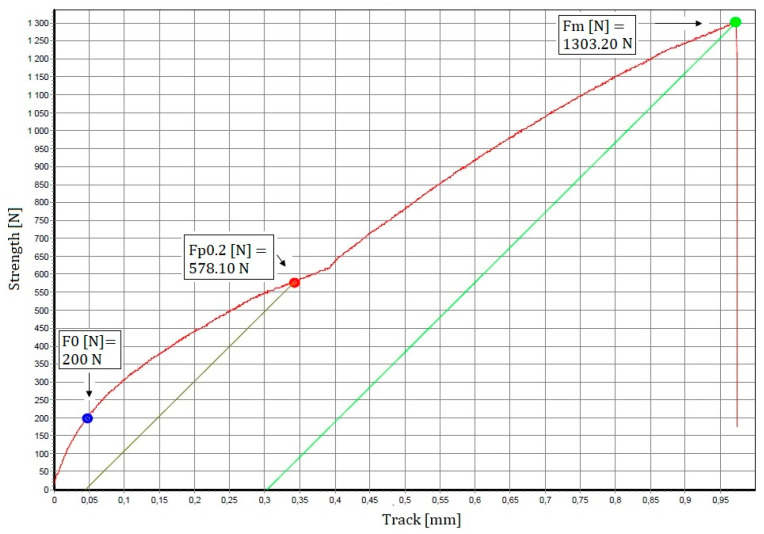
Tensile strength curve of an experimental solder alloy.

**Figure 14 materials-16-02991-f014:**
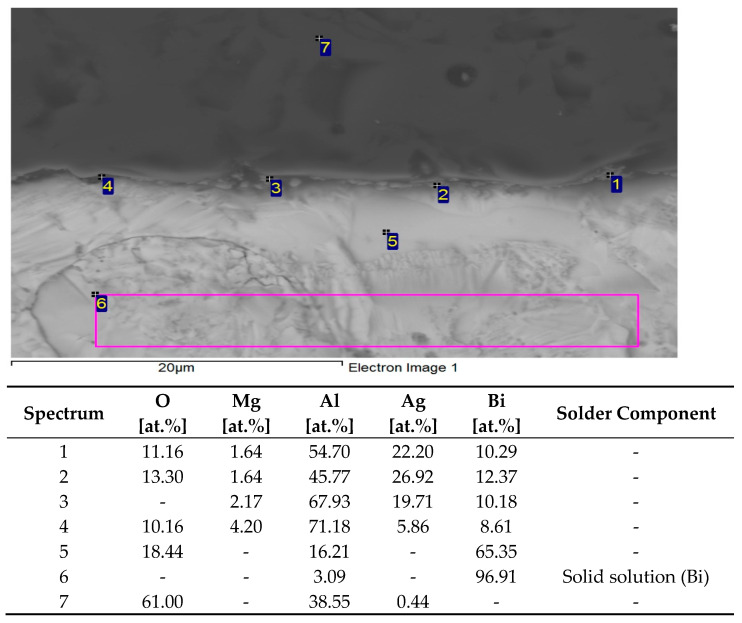
EDX point analysis of the Al_2_O_3_/Bi11Ag1Mg joint.

**Figure 15 materials-16-02991-f015:**
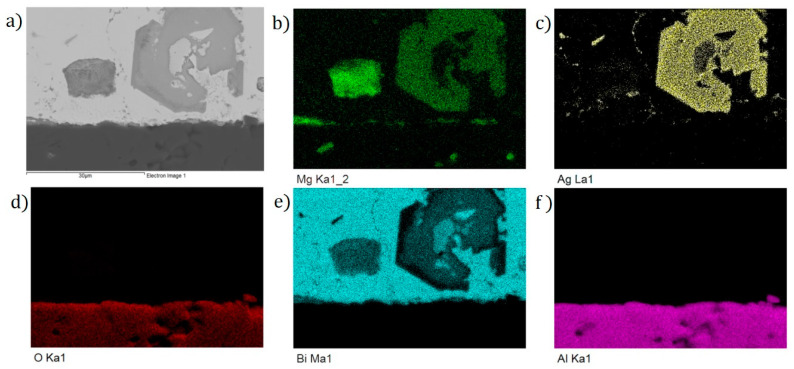
Planar distribution of elements on Al_2_O_3_/Bi11Ag1Mg joint boundaries: (**a**) joint microstructure; (**b**) Mg; (**c**) Ag; (**d**) Bi; (**e**) Al; and (**f**) O.

**Figure 16 materials-16-02991-f016:**
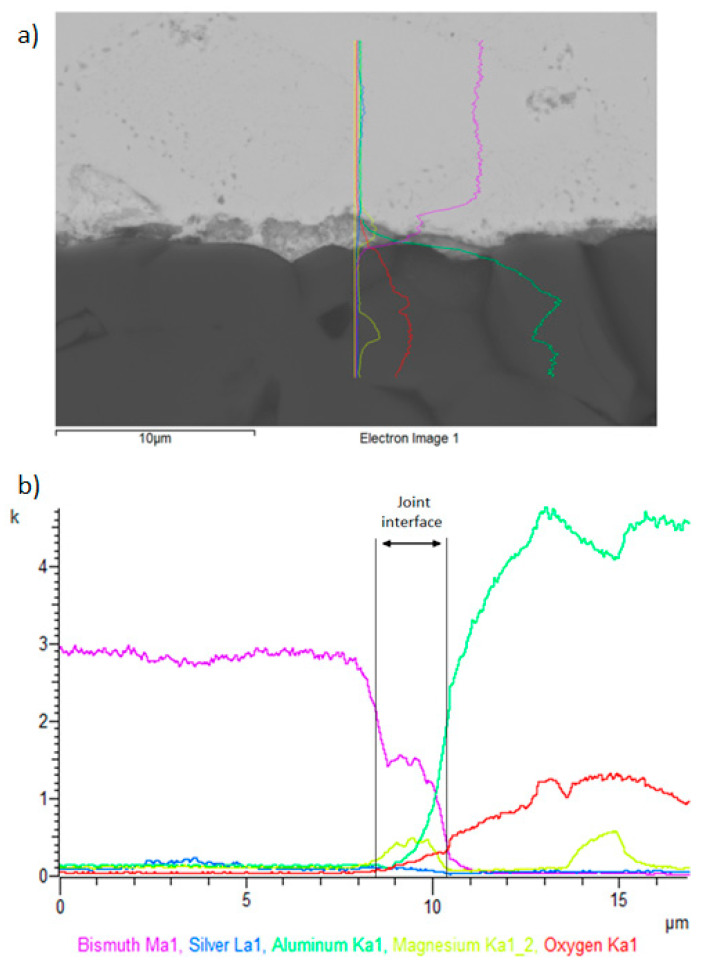
Line EDX analysis of the Al_2_O_3_/Bi11Ag1Mg joint: (**a**) transition zone with a marked line; (**b**) concentration profiles of Bi, Ag, Mg, Al, and O elements.

**Figure 17 materials-16-02991-f017:**
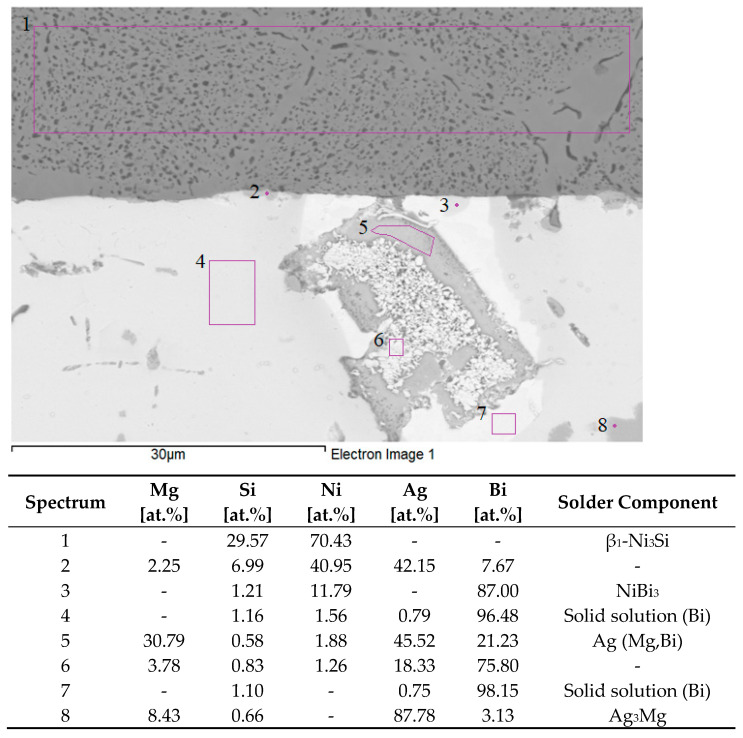
EDX analysis of boundary in Bi11Ag1Mg/Ni–SiC joint.

**Figure 18 materials-16-02991-f018:**
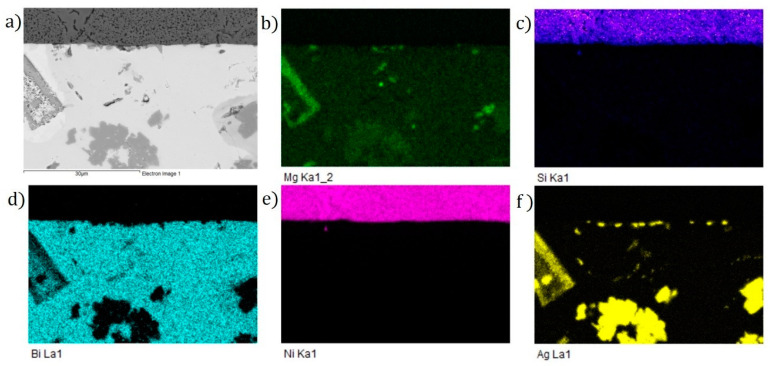
Planar distribution of elements at the boundary of the Bi11Ag1Mg/Ni–SiC joint: (**a**) joint microstructure; (**b**) Mg; (**c**) Si; (**d**) Bi; (**e**) Ni; and (**f**) Ag.

**Figure 19 materials-16-02991-f019:**
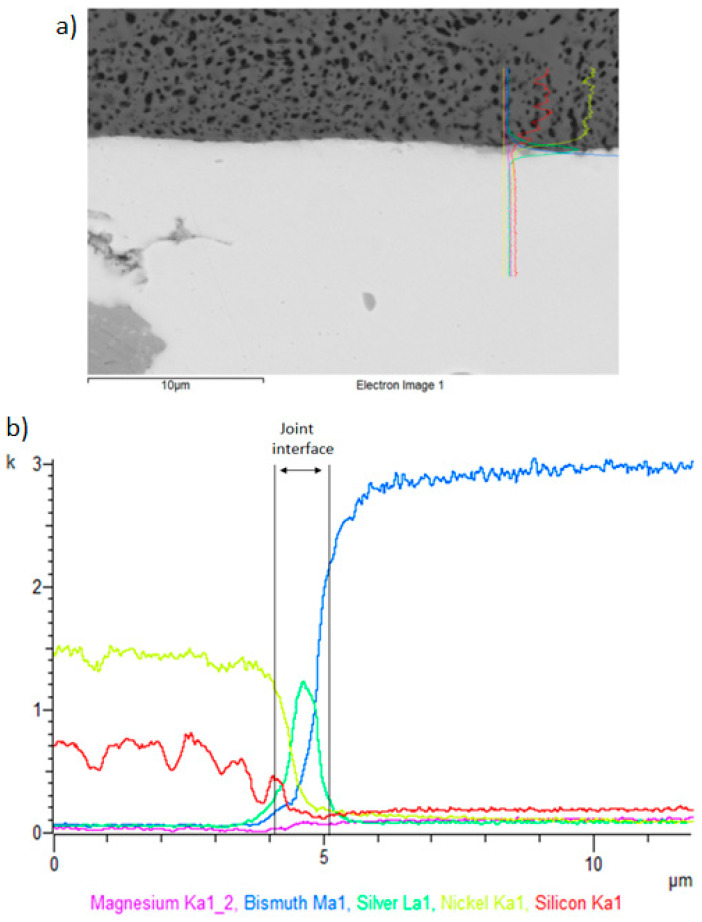
Line EDX analysis for the Bi11Ag1Mg/Ni–Sic joint: (**a**) a marked transition zone; (**b**) concentration profiles of Mg, Bi, Ag, Ni, and Si elements.

**Figure 20 materials-16-02991-f020:**
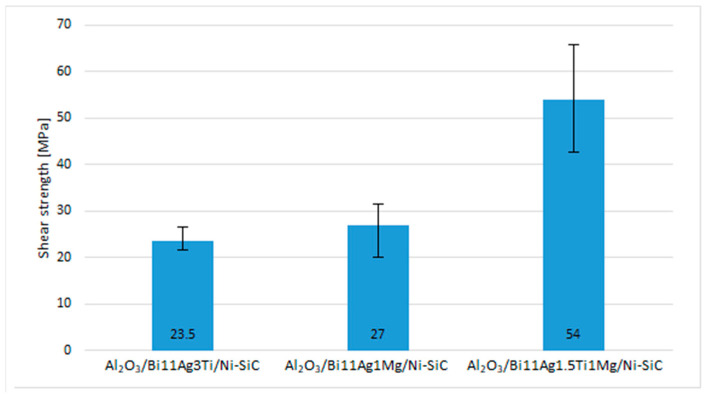
Shear strength measurement of soldered joints.

**Figure 21 materials-16-02991-f021:**
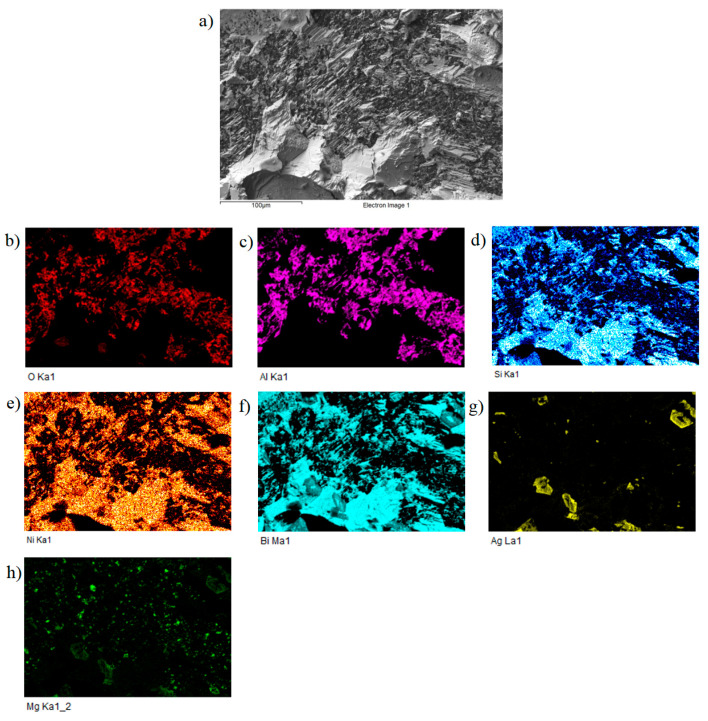
Fractured surface of the soldered joint of Al_2_O_3_/Bi11Ag1Mg/Ni–SiC and the planar distribution of individual elements: (**a**) fracture morphology; (**b**) O; (**c**) Al; (**d**) Si; (**e**) Ni; (**f**) Bi; (**g**) Ag; and (**h**) Mg.

**Table 1 materials-16-02991-t001:** Composition of the Bi–Ag–Mg alloy and the results of the chemical analysis performed by the ICP-AES method [wt.%].

Sample	Charge [wt.%]			ICP-AES [wt.%]		
	Bi	Ag	Mg	Bi	Ag	Mg
Bi11Ag1Mg	88.0	11.0	1.0	86.6	12.2 ± 0.6	1.18 ± 0.6

**Table 2 materials-16-02991-t002:** Soldering parameters.

Ultrasound power	400 W
Work frequency	40 kHz
Amplitude	2 μm
Soldering temperature	380 °C
Time of ultrasound acting	5 s

**Table 3 materials-16-02991-t003:** Significant phase transformation temperatures were determined by TG/DTA analysis.

Bi11Ag1Mg	Onset Point 1	Onset Point 2	T_L_ (°C)	T_E_ (°C)
Heating	264	-	380	263
	262	-	380	-
Cooling	243	340	357	-
	244	332	355	-

T_L_—liquidus temperature; T_E_—temperature of eutectic transformation.

## Data Availability

The data presented in this study are available upon request from the appropriate author.

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
