# Peer review of "Study of Bond Formation in Ceramic and Composite Materials Ultrasonically Soldered with Bi–Ag–Mg-Type Solder"

_materials, 2023, doi:10.3390/ma16082991_

Round 1

Reviewer 1 Report

This paper investigates the bond formation in a Bi-Ag-Mg soldering alloy and the direct soldering of Al2O3 ceramics and Ni-SiC composite. The work is very interesting and written in a fluent language. For this reason, it is recommended that the publication could be accepted after minor revision.

1)      In Introduction section, the number of references should be increased with novel references.

2) There are many grammatical errors in the manuscript. I suggest the author take help from his native English teachers to polish the manuscript. 

3) Conclusions, as they are now, look mostly like observations. These must be rewritten to reflect the novel outcomes of the work.

I suggest a minor revision for this paper.

Author Response

1) Answer by: In the article we have expanded the introduction by a few references. 
2) Answer by:  Our article has been sent for English proofreading. We are enclosing a certificate of this proofreading.
3) Answer by:  In this article and our previous articles we write conclusions in bullet points. The conclusion has been rewritten and completed.

Reviewer 2 Report

1. This paper introduces the study of a Bi-Ag-Mg soldering alloy and the direct soldering of Al2O3 ceramics and Ni-SiC composite. 2. The research results are original to some extent, but I personally think they have no significant contribution to this field. 3, In Figure 9, the size of Ag (Mg+Bi) phase is relatively large (>20um), and the distribution is uneven. Whether the microstructure can be controlled by changing the brazing process, the phase size can be small and uniformly distributed in the matrix, and the mechanical property of the solder layer can be further improved 4. The conclusion is consistent with the evidence and arguments put forward. In addition, 1, Line185-187, which is the writing template text, is recommended to be deleted. 2, Lines 242 and 398. "The phase diagram of Ag-Bi-Mg system is not available." In fact, this article does not involve the Ag-Bi-Mg ternary phase diagram, so it is recommended to delete it.

Author Response

1. Answer by: Thank you, you for your opinion.
2. Answer by: Thank you for your opinion.
3. Answer by: Yes, the soldering process can affect the microstructure and mechanical properties of the solder layer. Temperature changes and rapid cooling can occur during soldering, which can cause the formation of different phases and microstructures in the solder layer. 
4. Answer by: Those minor edits you suggested in the text have been checked and deleted.

Reviewer 3 Report

The manuscript titled “Study of bond formation in ceramic and composite materials ultrasonically soldered with Bi-Ag-Mg-type solder” is well written and can be considered for publication after addressing the following points.

·        Why Bi-Ag-Mg alloy was selected as a soldering alloy in the present study?

·        How many samples were considered for the tensile test and shown the standard deviations? Authors should provide the tensile stress-strain curve for a better understanding of the potential readers.

·        Authors need to show the dimension of the shear strength test sample or which standard was followed to measure the shear strength of the materials including the name of the experimental setup. 

Author Response

  1. Answer by: This Bi11Ag1Mg type alloy was selected as an alternative replacement for high lead solder. In order to make this Bi11Ag type alloy an active solder, only 1 wt. % Mg was added to it, which reacts at the interface from the Al2O3 ceramic and the Ni-SiC substrate.

  1. Answer by: The text on page 9 gives both the number of specimens and their ductility. For the tensile test, 3 specimens were used. For better understanding, we have added the tensile stress strain curve of one specimen in the article.
  2. Answer by: The dimensions of this specimen are given in the text on page 3 and in Figure 2b. The standard used was STN 05 0044. In our study, a special preparation was experimentally designed by us, which has already been published in several of our publications. I am presenting the internet links of our articles where we also used the same special preparation:

https://www.mdpi.com/2075-4701/11/4/624 https://www.aimspress.com/article/doi/10.3934/matersci.2023012 https://pubmed.ncbi.nlm.nih.gov/35955233/

Reviewer 4 Report

The manuscript shows interesting results of the study of the ultrasonic soldering of composites an ceramic sustrates, but some questions an observations are bellow.

- Indicate or specify units in figures 1 and 2.

- As shematic representation of the experiments, there can be suggested for authors an scale bar for figures 3 to 5.

- How authors determine the soldering parameters? How authors made constant the operating variables during soldering?

- How alloying elements contribute to the precipitation of the observed phases and the amount of it's precipitates, size and distribution in the matrix?

- In the ternary alloy, how possible is the formation of a ternary phase? If it is possible, How can this affect the mechanical properties?

- Explain in the conclusions the importance of the elemental composition used in this process according to other soldering processes

Author Response

1. Answer by: All these values are given in the text preceding the figures. These units are given in millimetres.
2. Answer by: Thank you for the suggestion you have provided we will consider this suggestion. 
3. Answer by: We determine the soldering parameters based on our previous published experiments and the melting temperature of the solder alloy. The melting temperature of the solder alloy is determined based on DTA analysis. We ensured the constancy during brazing by accurately measuring the brazing temperature using a NiCr/NiSi thermocouple.
4. Answer by: Overall, alloying elements can contribute to the formation and distribution of precipitates in the matrix in a variety of ways and their influence can be very complex. Therefore, it is important to know the properties and characteristics of alloying elements and their interactions with the matrix and with other alloy elements in order to properly design and fabricate alloys with the desired mechanical properties. In the Bi-Mg system there is a stable intermetallic phase α-Mg3Bi2 and in the Ag-Mg system there is an AgMg phase in a relatively wide range of concentrations. Also, high Bi and Ni contents were found in the experiments which may indicate the existence of NiBi3 phase.
5. Answer by: The formation of a ternary phase in a ternary alloy can be influenced by various factors such as alloy composition, temperature and heat treatment time. These factors can influence the dissolution and precipitation of individual elements in the alloy and the formation of new phases. We do not have such a ternary Bi-Ag-Mg diagram available we only have binary diagrams of Bi-Ag, Bi-Mg and Ag-Mg. 
6. Answer by: This Bi11Ag1Mg type alloy was selected as an alternative replacement for high lead solder. In order to make this Bi11Ag type alloy a permanent active solder, only 1 wt. % Mg was added to it, which reacts at the interface from the Al2O3 ceramic and the Ni-SiC substrate. Bi us in the solder provides a reduction in melting temperature and also increases wettability. Ag in the solder increases the mechanical strength and also improves the electrical conductivity.

Round 2

Reviewer 3 Report

The manuscript can be accepted in present form.